# Flexible and Highly Sensitive Hydrogen Sensor Based on Organic Nanofibers Decorated by Pd Nanoparticles

**DOI:** 10.3390/s19061290

**Published:** 2019-03-14

**Authors:** Hongchuan Jiang, Yibing Yu, Luying Zhang, Jun Zhu, Xiaohui Zhao, Wanli Zhang

**Affiliations:** State Key Laboratory of Electronic Thin Films and Integrated Devices, University of Electronic Science and Technology of China, Chengdu 610054, China; uestc_ybyu@163.com (Y.Y.); luyingz@163.com (L.Z.); junzhu@uestc.edu.cn (J.Z.); wlzhang@uestc.edu.cn (W.Z.)

**Keywords:** hydrogen sensor, organic nanofibers, high sensitivity, sputtering, nanoparticles

## Abstract

A highly sensitive and flexible hydrogen sensor based on organic nanofibers decorated by Pd nanoparticles (NPs) was designed and fabricated for low-concentration hydrogen detection. Pd NPs were deposited on organic nanofiber materials by DC magnetron sputtering. The temperature dependence of the sensitivity at 25 ppm H_2_ was characterized and discussed, and the maximum response of the sensor increased linearly with increasing measurement temperature. Performances of the hydrogen sensor were investigated with hydrogen concentration ranging from 5 ppm to 50 ppm. This sensor exhibits high sensitivity, with the response up to 6.55% for H_2_ as low as 5 ppm, and the output response of the hydrogen sensor increased linearly with the square root of hydrogen concentration. A cycling test between pure nitrogen and 25 ppm hydrogen concentration was performed, and the hydrogen sensor exhibited excellent consistency.

## 1. Introduction

Hydrogen gas is one of the most promising cleaning energy sources due to its high combustion efficiency and zero pollution to the environment. Meanwhile, hydrogen is also widely employed in the aerospace industry, metallurgical industry, and other fields [1,2,3]. However, it has a great security risk in practical applications since it is colorless, odorless, and explosive at concentrations above 4% in atmosphere [4,5]. Therefore, highly sensitive hydrogen sensors are urgently demanded. In particular, low concentration hydrogen detection is of great importance for the space exploration. By analyzing the space distribution of the hydrogen element, the formation and evolution of the universe, the impact of the space environment on the human living environment, and the origin of life could be evaluated and studied [6,7]. Nevertheless, the density of hydrogen in the space is extremely low, with varied temperature. So far, the hydrogen sensor with high sensitivity and repeatability, especially in the low H_2_ concentration range, is still a challenge.

Generally, the hydrogen sensors can be classified into electrochemical, electrical, and optical types [8]. The electrical sensor, as represented by metal-oxide or metal film-based sensors, are most promising due to their advantages such as simple structure, low power consumption, and fast response [8]. However, the operating temperature of the metal oxide-based sensors is usually above 100 ℃, and the sensitivity of the sensor drops dramatically with the decreasing operating temperature [9,10]. In addition, metal oxides respond to reducing gases such as H_2_, H_2_S, and CO. In order to overcome the shortcomings of poor hydrogen selectivity, new methods have to be adopted [11]. In contrast, the metal film-based sensors could work at low temperature; however, the performance of the sensor at low H_2_ concentration is still unsatisfied [12,13,14]. In order to improve the performance of the sensor, nanomaterials with large specific surface areas have been widely introduced into the hydrogen sensor [15,16,17,18].

In this paper, a low H_2_ concentration sensor based on an organic nanofiber system was designed and fabricated. The introduction of this nanosystem could increase the specific surface area and improve the hydrogen response performance. Pd nanoparticles (NPs) were deposited on the organic nanofibers by the sputtering technology, which is a fascinating method with convenience for the fabrication of hydrogen sensors. The performances of the hydrogen sensor under different H_2_ concentrations and operating temperatures were characterized, and the repeatability of the sensor was tested and discussed.

## 2. Experimental

### 2.1. Depositation of Pd NPs

The main component of nanofibers is polyacrylonitrile, which was fabricated on alumina foil by the electrospin method. Pd NPs were deposited on the organic nanofibers by direct current (DC) magnetron sputtering technology. High-purity Pd metal was used as the target, and the distance between the target and substrate was fixed at 110 mm. The background pressure was 8 × 10^−4^ Pa, and sputtering pressure was 0.4 Pa. Pd NPs were deposited on the nanofibers by 180 W, and the sputtering time was 45 s.

The morphology and composition of Pd NPs/organic nanofibers were examined with scanning electron microscopy (SEM, Inspect-F) and energy dispersive X-ray (EDX) mapping. The crystal structure of the sample was identified with X-ray diffraction (XRD, DX-1000).

### 2.2. Fabrication of the Sensor

This nanostructure composed of Pd NPs/organic nanofibers was cut to the size of 2 × 1 cm and transferred onto the flexible printed circuit board (PCB). Silver paste was applied on both sides of the synthesized nanostructure as the electrodes, and the distance between the electrodes was about 10 mm. The whole fabrication process of the sensor is schematically illustrated in Figure 1, and the image of the fabricated hydrogen sensor is shown in Figure 2.

### 2.3. Hydrogen Sensing Measurement

The measurement of the hydrogen sensor was carried out with a self-developed measurement system, as shown in Figure 3. This system is composed of mass flow controllers (MFC), a gas mix chamber, a gas test chamber, a temperature controller, and a Keithley 2400 Source Meter. The hydrogen sensor was put in the test chamber at atmospheric pressure, and the test chamber was placed in the oven to control the temperature of the measurement. Before the measurement, the oven was heated up to the desired temperature, and the chamber was purged with 99.999% pure nitrogen for 1 h. After the base resistance of the sensor (the resistance of the sensor under nitrogen) was measured, both nitrogen and 50 ppm hydrogen were introduced through the MFC into the gas mix chamber and mixed. The flow rates of the two gases were automatically controlled by gas distribution software on the computer to obtain H_2_ in different concentrations. After mixing, the gas mixture was delivered to the test chamber with a constant flow rate of 80 sccm. The resistance of the hydrogen sensor was acquired by a four-terminal method using a LabVIEW program (National Instruments) through a Keithley 2400 Source Meter under constant current of 1 mA. The output response (Rs) is defined as following:(1)$$R_{S}\left(\%\right)=\frac{R_{o}-R_{H}}{R_{o}}\times 100=\frac{\Delta R}{R_{o}}\times 100$$
where R_o_ and R_H_ were the base resistance and the resistance in the gas of interest, respectively [19].

## 3. Results and Discussion

Figure 4 is the SEM image of the synthesized nanostructure before and after the hydrogen test. The Pd NPs/organic nanofibers are straight with the diameter of 200–300 nm, and they were randomly oriented and overlapped with each other. The morphology of the synthesized nanostructure is well kept after the test, except that the surface of the nanofibers was rougher after the hydrogen test.

The EDX mapping result of the synthesized nanostructure was shown in Figure 5a. The area enclosed by the white rectangle was scanned, and the result was shown in the lower right corner, where the yellow dots represented the palladium element. As can be seen from the picture, a large number of Pd nanoparticles were concentrated in the middle of the scan image, corresponding to the position of the nanofiber in the rectangular region. Furthermore, only a small fraction of Pd nanoparticles appeared in the grayscale region where no nanofibers were present. The accumulation of Pd NPs on the surface of the organic nanofibers could be identified. As we can see, the Pd NPs were dispersed with each other, and no agglomeration occurred. The X-ray diffraction pattern of the synthesized nanostructure was presented in Figure 5b; only four strong diffraction peaks at 2θ values of 40.48°, 47.42°, 68.68°, and 82.92° were identified, which represent the (111), (200), (220), and (311) crystalline planes of Pd NPs, respectively. This result indicates that the Pd NPs are fully crystallized, and the crystal size of Pd (calculated with the Scherrer equation) is about 22 nm.

Figure 6a shows the electrical resistance of the hydrogen sensor as a function of time when exposed to 5 ppm H_2_. The resistance drops quickly in the initial period of contact with hydrogen at 70 ℃ until the equilibrium is reached. The schematic of the hydrogen sensor was shown in Figure 6b. The Pd NPs were deposited evenly on the organic nanofibers with magnetron sputtering. Since the insulating organic nanofibers are randomly overlapped with each other, the electrically conductive pathways might be formed in the synthesized nanostructure after the deposition of highly conductive Pd NPs. When the hydrogen was introduced, the Pd NPs could react with H_2_ and form a Pd–H compound:(2)$$\frac{1}{2}H_{2}+{\mathrm{Pd}}_{\mathrm{active}}\overset{k_{1}}{⇄}\mathrm{PdH}$$
where Pd_active_ is the active sites of Pd NPs for hydrogen reaction. This reaction also leads to the lattice expansion of the conductive Pd NPs [20], which is evidenced by the rougher surface of the nanofibers, as demonstrated in the aforementioned SEM results. As a result, the distance between adjacent NPs is reduced; the initially isolated Pd NPs could come into contact with each other, and the conductive pathway with a shorter distance significantly increased [21]. This process is highly sensitive even under low H_2_ concentrations, and results in the substantial reduction of the resistance of the synthesized nanostructure, which well explains the high sensitivity of the sensor.

Figure 7a shows the response of the hydrogen sensor exposed to 25 ppm hydrogen under different test temperatures. After the inlet of the H_2_, the resistance of the synthesized nanostructure decreases rapidly during the initial few minutes, and reaches the equilibrium after about 30 min. The similar results of response and recover time have been reported previously in a Pd/reduced graphene oxide-based sensor [22]. However, the sensor in our work exhibited higher sensitivity and detection, with the response up to 12.2% for 25 ppm H_2_ [13,22]. Figure 7b shows the temperature dependence of the maximum response of the hydrogen sensor exposed to 25 ppm H_2_. In sharp contrast to a previous report [22], the maximum response of the sensor increased almost linearly with the increasing test temperature. It is believed that the chemical adsorption of the hydrogen, which is more dependent on the driving force other than the physical adsorption, could be significantly facilitated with increasing temperature and lead to metal hydride formation with volume expansion [20,23]. Although the formation of the hydride would lead to the increase of the resistance, it is subordinate compared with the increment and shortening of the conduction pathway inside the synthesized nanostructure. The synergistic effects lead to evident decrement of the resistance of the synthesized nanostructure with increasing measurement temperature.

Figure 8a shows the response of the hydrogen sensor under different H_2_ concentrations from 5 ppm to 50 ppm. The response of the synthesized nanostructure increases quickly during the first dozen of minutes of the exposure to H_2_, and reaches saturation after about 15 min. Since nitrogen was used as the desorption gas, the recovery of the sensor is relatively slow [20,23]. The linear correlation between the response and the square root of H_2_ concentration shown in Figure 8b might be related to the adsorption process of Pd NPs according to the Sievert’s law [24].
(3)$${\left(\frac{H}{Pd}\right)}_{at}=K_{S}\times \sqrt{p_{H_{2}}}$$
where (H/Pd)_at_ is the atom ratio of H and Pd components, K_s_ is the Sievert’s constant, and P_H_2__ is the H_2_ partial pressure in the environment. When the maximum response of the sensor is achieved, the reaction for hydrogen sensing (Equation (2)) reaches the equilibrium, and the following relationship can be easily deduced from equations (2) and (3):(4)$$\left[\mathrm{PdH}\right]=k_{1}{\left[H_{2}\right]}^{\frac{1}{2}}\cdot {\mathrm{Pd}}_{\mathrm{active}}$$
where [PdH] and [H_2_] is the amount of PdH compound and the concentration of hydrogen, respectively. When the hydrogen concentration is low, there are plenty of Pd active sites, and the concentration of them almost remains constant before and after the hydrogen sensing process. As a result, the amount of PdH compound at low hydrogen concentration in inert gas is linearly related with the square root of the hydrogen concentration, which sufficiently explains the experimental results in Figure 8b [13,20].

A cycling test between pure nitrogen and 25 ppm hydrogen was performed at 343 K, and the result is shown in Figure 9. The response of the sensor increased rapidly after the inlet of the hydrogen, while it decreased slowly after purging with nitrogen. The maximum response of the sensor only demonstrates a moderate decrement with increasing measurement cycles. This could be ascribed to the not fully desorption of PdH. Since the organic nanofibers were randomly oriented and overlapped with each other, a few Pd–H compounds might exist in the accumulation place even after the introduction of nitrogen. However, it should be noted that even after three measurement cycles, the sensor still maintains high sensitivity to low concentrations of hydrogen with the maximum response of 10.63%, indicating the excellent sensitivity of the sensor [25,26].

## 4. Conclusions

A flexible and highly sensitive hydrogen sensor based on organic nanofibers was designed and fabricated. Compared with a conventional electrical hydrogen sensor, the fabricated sensor exhibits superior sensitivity to hydrogen with concentration as low as 5 ppm over a wide temperature range. Since the chemical absorption is facilitated with increasing temperature, the maximum response of the sensor increased linearly with the increasing measurement temperature, and the maximum response of the sensor increased linearly with the square root of the hydrogen concentration. This hydrogen sensor also demonstrates excellent durability under low hydrogen concentration over multiple measurement cycles. Overall, this nanofiber-based hydrogen sensor demonstrates great potential for low-concentration hydrogen detection in aerospace exploration.

## Figures and Tables

**Figure 1 sensors-19-01290-f001:**
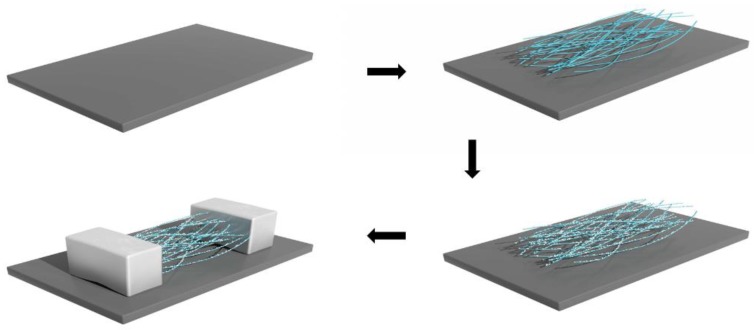
Schematic diagram of the hydrogen sensors.

**Figure 2 sensors-19-01290-f002:**
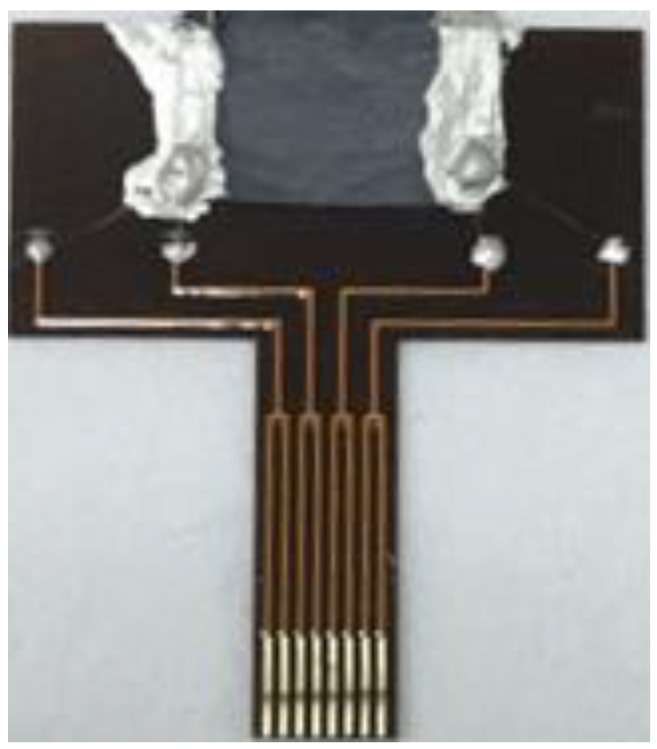
The fabricated hydrogen sensor.

**Figure 3 sensors-19-01290-f003:**
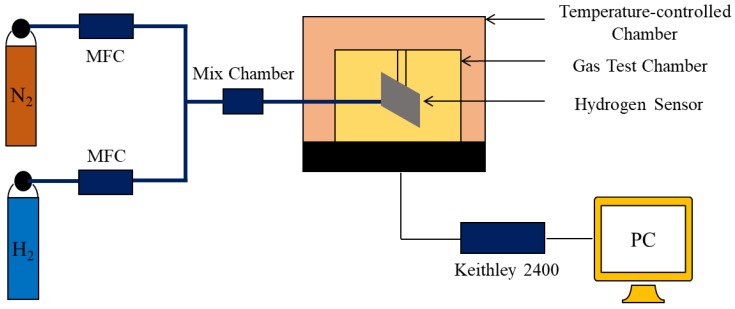
Schematic of the hydrogen measuring system.

**Figure 4 sensors-19-01290-f004:**
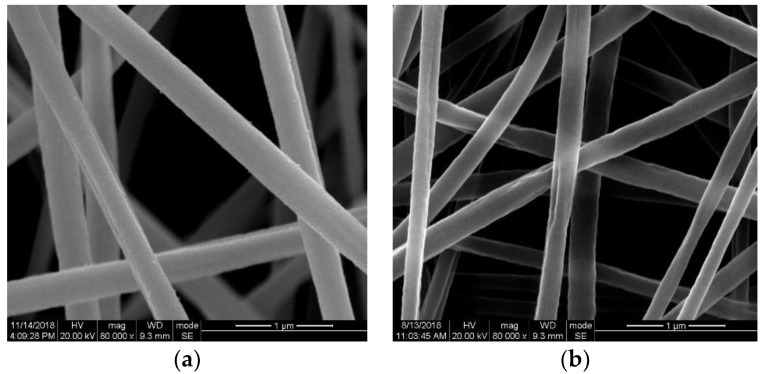
SEM image of the synthesized nanostructure (**a**) before and (**b**) after the H_2_ test.

**Figure 5 sensors-19-01290-f005:**
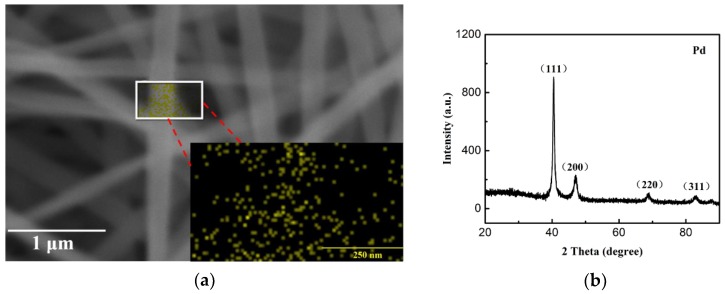
(**a**) Energy dispersive X-ray (EDX) mapping of the synthesized nanostructure, and (**b**) X-ray diffraction pattern of the synthesized nanostructure.

**Figure 6 sensors-19-01290-f006:**
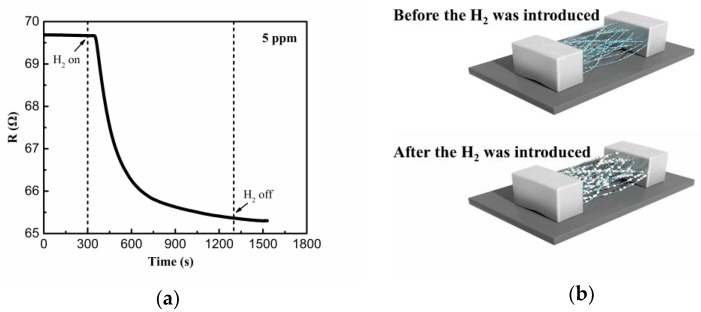
(**a**) Electrical resistance changes of the hydrogen sensor when exposed to 5 ppm H_2_. (**b**) The schematic of the hydrogen sensor before and after H_2_ exposure.

**Figure 7 sensors-19-01290-f007:**
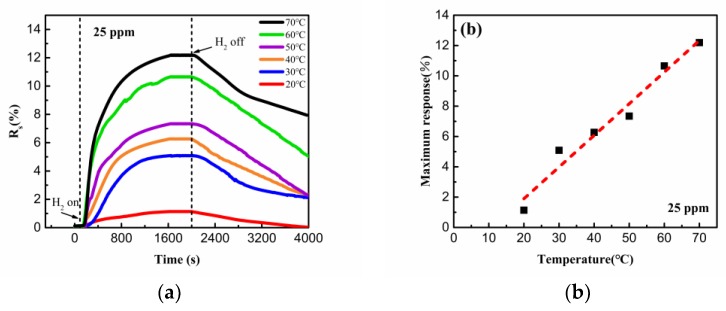
(**a**) The response of the sensor under 25 ppm H_2_ at different temperatures. (**b**) The maximum response as a function of measurement temperature.

**Figure 8 sensors-19-01290-f008:**
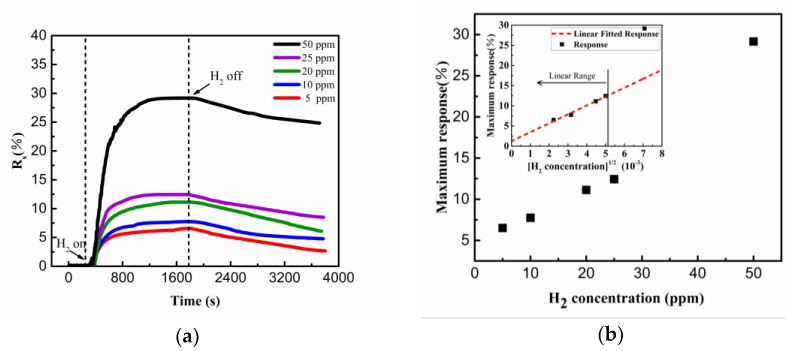
(**a**) The response of the sensor under different H_2_ concentrations at 70 ℃. (**b**) The maximum response as a function of square root of H_2_ concentration.

**Figure 9 sensors-19-01290-f009:**
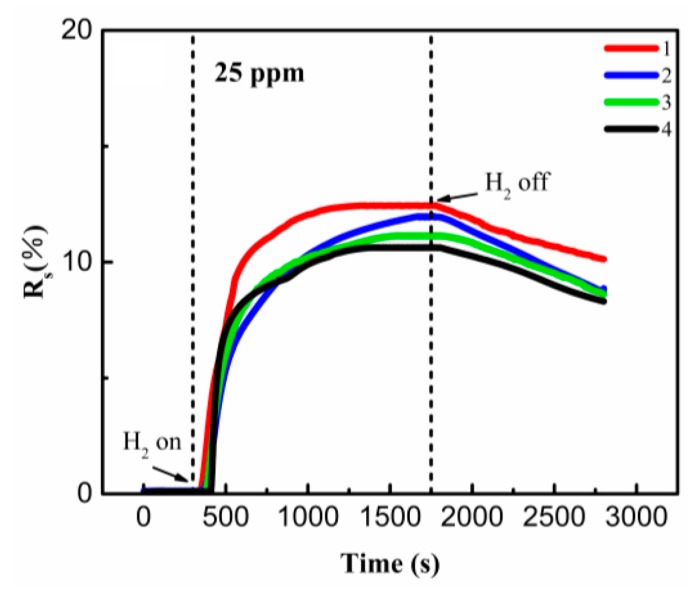
The response of the sensor under 25 ppm H_2_ in nitrogen.

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
