# Peer review of "Flexible and Highly Sensitive Hydrogen Sensor Based on Organic Nanofibers Decorated by Pd Nanoparticles"

_sensors, 2019, doi:10.3390/s19061290_

Round 1
Reviewer 1 Report
This is interesting paper but requires some additional information before being published:
Fig. 2, you haven't add any information about the distance between the electodes of the applied substrate. This is very important for further development of the presented sensor.
The reported DC rsistance is rather low. How could you increase it, e.g., by changing distance between the electrodes or by
What is the reason in various response time for different gas concentrations? Please explain it.
Response time is rather long. How could you decrease it. Please give some discussion about introducing additives improving the gas sensing resonse or applying UV light (I suppose it should work for this material). Another possibile discussion would be application of other dynamic modes og gas sensor operation. Please introduce some discussion in the manuscript about exemplary dynamic mode, like flicker noise measurements: Kotarski, M., & Smulko, J. (2009). Noise measurement set-ups for fluctuations-enhanced gas sensing. Metrol. Meas. Syst, 16(3), 457-464.
OX axes in the Fig. 7b requires amendment. You have to introduce the space there.
Fig. 8b - the inner figure is invisible. I can't read the details. You have to improve it.
Author Response
Response to Reviewer,
Thank you very much for spending time to carefully review our work and give us thoughtful suggestions. These instructive comments stimulate the further improvement of our manuscript. Accordingly, we have suitably addressed these comments in the revised manuscript. Our responses are stated as below.
Q1: Fig. 2, you haven't added any information about the distance between the electrodes of the applied substrate. This is very important for further development of the presented sensor.
Response: Thank you very much for the kind suggestion. We have added the information about the distance between the electrodes of the applied substrate in revised manuscript. “Silver paste was applied on both sides of the synthesized nanostructure as the electrodes, the distance between the electrodes is about 10mm.”
Q2: The reported DC resistance is rather low. How could you increase it, e.g., by changing distance between the electrodes?
Response: It’s true that the resistance of the sensor is rather low from the metal nanoparticles (NPs). As mentioned in the experimental section, Pd NPs were deposited by DC magnetron sputtering technology and the sputter power was 180 W, the sputtering time was 45 s. The resistance of the device can be properly increased by reducing the sputtering power and shortening the sputtering time for the reduced particle size. Meanwhile, increasing the distance between the silver paste applied at both sides is also efficient for increasing the resistance of the sensor.
Q3: What is the reason in various response time for different gas concentrations? Please explain it.
Response: The Pd NPs could react with H2 and form Pd-H compound:
And the adsorption process of Pd NPs follows the Sievert’s law:
When the temperature is constant, the solubility of H in Pd increases with the increased partial pressure of H2. In consequence, the higher of the hydrogen partial pressure, the faster generation rate of Pd-H compound, because the positive chemical reaction is promoted. As a result, the response time of the sensor varies for different gas concentrations.
Q4: Response time is rather long. How could you decrease it? Please give some discussion about introducing additives improving the gas sensing response or applying UV light (I suppose it should work for this material). Another possible discussion would be application of other dynamic modes of gas sensor operation. Please introduce some discussion in the manuscript about exemplary dynamic mode, like flicker noise measurements: Kotarski, M., & Smulko, J. (2009). Noise measurement set-ups for fluctuations-enhanced gas sensing. Metrol. Meas. Syst, 16(3), 457-464.
Response: Thank you for your critical question on our work. As discussed above, the response time is highly influenced by the concentration of H2. For the optimization of response time, we can introduce novel structures, such as graphene/Pd NPs structure (Hong, J.; Lee, S.; Seo, J.; Pyo, S.; Kim, J.; Lee, T. A highly sensitive hydrogen sensor with gas selectivity using a PMMA Membrane-coated Pd nanoparticle/single-layer graphene hybrid. ACS Appl. Mater Interfaces 2015, 7, 3554-3561.).
The introduction of UV light is positive for the synthesis of PdH in principle. However, as limited by our homemade measurement system, we cannot introduce UV light during the test. As recommended, the advantages of exemplary dynamic modes have been discussed in the revised manuscript as follows. “In addition, metal oxides respond to reducing gases such as H2, H2S, and CO. In order to overcome the shortcomings of poor hydrogen selectivity, new methods have to be adopted [11].”
Q5: X axes in the Fig. 7b requires amendment. You have to introduce the space there.
Response: Thanks for bringing this issue to our attention. We have made corresponding changes to Fig.7b.
Q6: Fig. 8b - the inner figure is invisible. I can't read the details. You have to improve it.
Response: Thanks for your suggestion. We are very sorry for our negligence, we have revised Fig.8b and explained it in the manuscript. The detailed information are as below.
Figure. 8(b)
The linear correlation between the response and the square root of H2 concentration shown in Figure 8 (b) might be related to the adsorption process of Pd NPs according to the Sievert’s law.
(3)
Where (H/Pd)at is the atom ratio of H and Pd components, Ks is the Sievert’s constant, and PH2 is the H2 partial pressure in environment. When the maximum response of the sensor is achieved, the reaction for hydrogen sensing (equation 2) reaches the equilibrium.
(2)
And the following relationship can be easily deduced from equation (2) and equation (3):
(4)
Where [PdH] and [H2] is the amount of PdH compound and the concentration of hydrogen, respectively. When the hydrogen concentration is low, there are plenty of Pd active sites and the concentration of them almost keeps constant before and after the hydrogen sensing process. As a result, the amount of PdH compound at low hydrogen concentration in inert gas is linearly related with the square root of the hydrogen concentration, which well explains the experiment results in Figure 8 (b), and we can also see from Figure 8 (b) that the relationship between response and H2 concentration is exponential.

Reviewer 2 Report
This study describes the development of hydrogen (H2) sensor using organic nano fibers with palladium (Pd) nanoparticles. The authors demonstrated that Pd nanoparticles, deposited on the surface of nano fibers, form the electrical conductive pathway and allow the increment of maximum response with increasing temperature or H2 concentration due to chemical adsorption of H2 on Pd nanoparticles and expansion of lattice of Pd nanoparticle, thereby resulting in high sensitivity under low H2 concentration. The approach seems to be interesting, and overall experimental procedures and results are well organized and written. However, several points should be improved before publication. Thus, I recommend the publication of the presented manuscript in the journal after major revision.
Specific comments:
1. In page 3 line 85, equation 1 is seemed to be wrong. “RH-R0” may be converted into “R0-RH” due to the decrease of resistance after introduction of H2.
2. In page 4 line 96, the authors describe that the accumulation of Pd NPs on the surface of the organic nano fibers was identified according to Figure 5a. Despite a presented EDX mapping result, it is unclear whether Pd NPs were deposited on the surface of the organic nano fibers. Therefore, better presentation of Figure 5a is required.
3. In page 5 line 133, what do the “synergistic effects” indicate? The authors claim that evident decrement of the resistance is caused by chemical adsorption of the hydrogen, rather than physical adsorption. I think that there are no more additional reported factors to support the result in the manuscript.
4. More detailed discussion on high sensitivity under low H2 concentration is required. Especially, the authors give a description for only analysis of data in Figure 8. Interpretation and mechanism studies on the result in Figure 8 are strongly suggested.
5. Some typos should be corrected. For example, in page 1 line 14, “magneton” and in page 5 line 128 and 129, “absorption” should be changed into “magnetron” and “adsorption”, respectively.

Author Response
Response to Reviewer,
Thank you very much for spending time to carefully review our work and give us thoughtful suggestions. These instructive comments stimulate the further improvement of our manuscript. Accordingly, we have suitably addressed these comments in the revised manuscript. Our responses are stated as below.
Specific comments:
Q1: In page 3 line 85, equation 1 is seemed to be wrong. “RH-R0” may be converted into “R0-RH” due to the decrease of resistance after introduction of H2.
Response: Thank you for the kind indication. According correction has been made in our revised manuscript.
Q2: In page 4 line 96, the authors describe that the accumulation of Pd NPs on the surface of the organic nano fibers was identified according to Figure 5a. Despite a presented EDX mapping result, it is unclear whether Pd NPs were deposited on the surface of the organic nano fibers. Therefore, better presentation of Figure 5a is required.
Response: Thank you for the insightful suggestion. As shown in Figure 1, a large number of Pd nanoparticles are visible on the surface of nano fibers, and only a few nanoparticles are distributed in the blanks. The distribution of Pd nanoparticles on as synthesized nanofibers can be demonstrated. Similar proofs have been reported in other published articles. (Hassan, K.; Uddin, A.I.; Chung, G.-S. Fast-response-hydrogen sensors based on discrete Pt/Pd bimetallic ultra-thin films. Sens. Actuators, B 2016, 234, 435-445).
Figure. 1
Q3: In page 5 line 133, what do the “synergistic effects” indicate? The authors claim that evident decrement of the resistance is caused by chemical adsorption of the hydrogen, rather than physical adsorption. I think that there are no more additional reported factors to support the result in the manuscript.
Response: As mentioned in manuscript, when the hydrogen was introduced, the Pd NPs could react with H2 and form Pd-H compound.
This reaction could lead to the lattice expansion, causing the increase of the resistance. At the same time, due to the volume expansion changing from Pd to Pd-H, the distance between adjacent NPs is reduced, the initially isolated Pd NPs could contact with each other and the conductive pathway with shorter distance is significantly increased. This will result in a significant reduction in resistance. “synergistic effects” refers to the joint effect of the two opposite effects on resistance. As indicated by
[20] Sun, Y.-G.; Wang, H.H.; Xia, M.-G. Single-walled carbon nanotubes modified with Pd nanoparticles: Unique building blocks for high-performance, flexible hydrogen sensors. J. Phys. Chem. C 2008, 112, 1250-1259;
[23] Ou, Y.-J.; Si, W.-W.; Yu, G.; Tang, L.-L.; Zhang, J.; Dong, Q.-Z. Nanostructures of Pd–Ni alloy deposited on carbon fibers for sensing hydrogen. J. Alloys Compd 2013, 569, 130-135.
The resistance mediation of Pd nanoparticles in hydrogen atmosphere originates from the chemical reaction process:
H2 ⇄ 2H
H + Pd ⇄ Pd-H
Q4: More detailed discussion on high sensitivity under low H2 concentration is required. Especially, the authors give a description for only analysis of data in Figure 8. Interpretation and mechanism studies on the result in Figure 8 are strongly suggested.
Response: Thanks for bringing this issue to our attention. We have made a detailed discussion about Figure 8 in the revised manuscript as required. The detailed information are as below.
Figure. 8 (b)
The linear correlation between the response and the square root of H2 concentration shown in Figure 8 (b) might be related to the adsorption process of Pd NPs according to the Sievert’s law:
(3)
Where (H/Pd)at is the atom ratio of H and Pd components, Ks is the Sievert’s constant, and PH2 is the H2 partial pressure in environment. When the maximum response of the sensor is achieved, the reaction for hydrogen sensing (equation 2) reaches the equilibrium.
(2)
And the following relationship can be easily deduced from equation (2) and equation (3):
(4)
Where [PdH] and [H2] is the amount of PdH compound and the concentration of hydrogen, respectively. When the hydrogen concentration is low, there are plenty of Pd active sites and the concentration of them almost keeps constant before and after the hydrogen sensing process. As a result, the amount of PdH compound at low hydrogen concentration in inert gas is linearly related with the square root of the hydrogen concentration, which well explains the experiment results in Figure 8 (b), and we can also see from Figure 8 (b) that the relationship between response and H2 concentration is exponential.
Q5: Some typos should be corrected. For example, in page 1 line 14, “magneton” and in page 5 line 128 and 129, “absorption” should be changed into “magnetron” and “adsorption”, respectively.
Response: Thanks for the suggestion. We are sorry about our mistake. Accordingly, all these typos have been checked carefully and corrected in our revised manuscript.

Reviewer 3 Report
This paper reports the fabrication of sensitive hydrogen sensors adopting organic fibers/Pd nanoparticles and their hydrogen sensing characteristics. The topic is of interest to researchers in the field, and the improvement of the sensitivity at low hydrogen concentration by using the nanostructure is considered an advance over previous works. However, the importance of this work is not fully recognized in its present form, so the quantitative comparison of sensitivity/detection limit with those reported in other published works is required to highlight the contribution of this paper. In addition, selectivity data should be added to prove the applicability of the proposed sensor. Finally, the paper contains some ambiguities that need to be addressed before a decision is made, as follows. (1) More detailed Experimental is needed. The concentration of H2 gas and the configuration /procedure of resistance measurement (4-terminal configuration?) should be specified. (2) Thinning of organic fibers after a test is observed in Fig.4(b). What is the reason? (3) Although there are the results of EDX mapping and XRD in the manuscript, they are indirect evidences to show Pd nanoparticle formation. Please provide TEM or SEM images of Pd nanoparticles. (4) The authors claim that the inset of Fig.8(b) represents the linear relationship between response and the square root of hydrogen concentration. However, the curve (response vs hydrogen concentration) in Fig.8(b) also seems to indicate that the response increase linearly with an increase of hydrogen concentration. Namely, it is unclear whether the response is proportional to hydrogen concentration or the square root of that. (5) More measurement cycles need to be performed to show its stability.
Author Response
Response to Reviewer,
Thank you very much for spending time to carefully review our work and give us thoughtful suggestions. These instructive comments stimulate the further improvement of our manuscript. Accordingly, we have suitably addressed these comments in the revised manuscript. Our responses are stated as below.
Q1: More detailed Experimental is needed. The concentration of H2 gas and the configuration /procedure of resistance measurement (4-terminal configuration?) should be specified.
Response: Thank you very much for the kind suggestion. We have added the related information in experimental section. The detailed information are as follows:
This system is composed of mass flow controllers (MFC), gas mix chamber, gas test chamber, temperature controller and Keithley 2400 Source Meter. The hydrogen sensor was put in the test chamber at atmospheric pressure, and the test chamber was placed in the oven to control the temperature of the measurement. Before the measurement, the oven was heated up to the desired temperature and the chamber was purged with pure nitrogen for 1h. After the base resistance of the sensor (the resistance of the sensor under nitrogen) was measured, both nitrogen and hydrogen were introduced through MFC into the gas mix chamber and mixed. The flow rates of the two gases were automatically controlled by Gas distribution software on the computer to obtain H2 with different concentration. After mixing, the gas mixture was delivered to the test chamber with a constant flow rate of 80 sccm. The resistance of the hydrogen sensor was acquired by a four-terminal method using a LabVIEW program (National Instruments) through a Keithley 2400 Source Meter under constant current of 1mA.
Q2: Thinning of organic fibers after a test is observed in Fig.4(b). What is the reason?
Response: It’s true that the organic fibers are thinner in Fig.4(b) than in Fig.4(a), but this is not due to the introduction of hydrogen during the test. The organic nano fibers are randomly oriented and overlapped with each other. And the uniformity of the material itself is very poor, we are unbale to control the same area selected of the two SEM images. Therefore, the thickness of the fibers in the two photos may be slightly different.
Q3: Although there are the results of EDX mapping and XRD in the manuscript, they are indirect evidences to show Pd nanoparticle formation. Please provide TEM or SEM images of Pd nanoparticles.
Response: Thank you for the insightful suggestion. As shown in Figure 1, a large number of Pd nanoparticles are visible on the surface of nano fibers, and only a few nanoparticles are distributed in the blanks. The distribution of Pd nanoparticles on as synthesized nanofibers can be demonstrated. Similar proofs have been reported in other published articles. (Hassan, K.; Uddin, A.I.; Chung, G.-S. Fast-response-hydrogen sensors based on discrete Pt/Pd bimetallic ultra-thin films. Sens. Actuators, B 2016, 234, 435-445).
Figure. 1
Q4: The authors claim that the inset of Fig.8(b) represents the linear relationship between response and the square root of hydrogen concentration. However, the curve (response vs hydrogen concentration) in Fig.8(b) also seems to indicate that the response increases linearly with an increase of hydrogen concentration. Namely, it is unclear whether the response is proportional to hydrogen concentration or the square root of that.
Response: Thank you for your critical question on our work. As indicated by
[13] Öztürk, S.; Kılınç, N. Pd thin films on flexible substrate for hydrogen sensor. J. Alloys Compd. 2016, 674, 179-184;
[20] Sun, Y.-G.; Wang, H.H.; Xia, M.-G. Single-walled carbon nanotubes modified with Pd nanoparticles: Unique building blocks for high-performance, flexible hydrogen sensors. J. Phys. Chem. C 2008, 112, 1250-1259.
The response is linear with the square root of hydrogen concentration. We are sorry that Fig.8(b) is confusing, and according correction and explanation have been made in our revised manuscript. The detailed information are as follows:
The linear correlation between the response and the square root of H2 concentration shown in Figure 8 (b) might be related to the adsorption process of Pd NPs according to the Sievert’s law [24].
(3)
Where (H/Pd)at is the atom ratio of H and Pd components, Ks is the Sievert’s constant, and PH2 is the H2 partial pressure in environment. When the maximum response of the sensor is achieved, the reaction for hydrogen sensing (equation 2) reaches the equilibrium.
(2)
And the following relationship can be easily deduced from equation (2) and equation (3):
(4)
Where [PdH] and [H2] is the amount of PdH compound and the concentration of hydrogen, respectively. When the hydrogen concentration is low, there are plenty of Pd active sites and the concentration of them almost keeps constant before and after the hydrogen sensing process. As a result, the amount of PdH compound at low hydrogen concentration in inert gas is linearly related with the square root of the hydrogen concentration, which well explains the experiment results in Figure 8 (b) [13, 20], and we can also see from Figure 8 (b) that the relationship between response and H2 concentration is exponential.
Q5: More measurement cycles need to be performed to show its stability.
Response: Thank you for the thoughtful suggestion. As shown in our manuscript, the maximum response of the sensor demonstrates moderate decrement with increasing measurement cycles, the stability of the device is not very good. This might be ascribed to the not fully desorption of PdH. Since the organic nano fibers were randomly oriented and overlapped with each other, there might existed a few Pd-H compound in the accumulation place even after the introduction of nitrogen. But it should be noted that even after three measurement cycles, the sensor still maintains high sensitivity to low concentration of hydrogen with the maximum response of 10.63%, indicating excellent sensitivity of the sensor. And how to improve the stability of the device is the focus of our next research.

Round 2
Reviewer 2 Report
Accept in present form
Author Response
Dear Reviewer,
Thank you so much for your reviewing! We deeply appreciate your recognition of our research work and appreciate the comment in improving the quality of this article.
Yours Sincerely
Hongchuan Jiang *, Yibing Yu

Reviewer 3 Report
The revised manuscript is improved compared with the previous version, but it still has some ambiguities that need to be addressed, as follows. (1) Selectivity data obtained using the fabricated device should be added to prove the applicability of the proposed sensor. (2) What is the concentration of hydrogen gas introduced into the gas mix chamber? If the gas is pure hydrogen, please provide its purity. (3) Figure. 1 in Response to Reviewer is still unclear so that nanoparticles can be found. It just shows yellow spots (maybe corresponding to the intensity from an element). Another clear image directly showing the nanoparticles is required. Otherwise, please explain more details about Figure. 1 and add it into the manuscript. (4) Regarding Fig.8(b), the authors claim that the relationship between response and hydrogen concentration is exponential. However, it is not compatible with the linear correlation between the response and the square root of hydrogen concentration. In addition, the curve (response vs hydrogen concentration) in Fig.8(b) should be concave downward.
Author Response
Response to Reviewer,
Thanks for spending time to carefully review our work and give us thoughtful suggestions. These instructive comments stimulate the further improvement of our manuscript. We have suitably addressed the comments and carefully revised the whole manuscript. Our responses are stated as below.
Specific comments:
(1) Selectivity data obtained using the fabricated device should be added to prove the applicability of the proposed sensor.
Response: Thank you for your kind suggestion. As indicated by
Hübert, T.; Boon-Brett, L.; Black, G.; Banach, U. Hydrogen sensors – A review. Sens. Actuators, B 2011, 157, 329-352.
Wiswall, R. Hydrogen storage in metals. Hydrogen in metals Ⅱ: Application-Oriented Properties. 1978, 29, 201-242.
“Hydrogen has a high solubility in palladium in particular and this interaction is also selective making palladium the metal of choice in the electrical type of sensor”. Unlike metal oxide types, metal Pd has high selectivity to hydrogen and the influence of other gases almost can be ignored. And this conclusion has been confirmed in other published articles. (Hassan, K.; Uddin, A.I.; Chung, G.-S. Fast-response hydrogen sensors based on discrete Pt/Pd bimetallic ultra-thin films. Sens. Actuators, B 2016, 234, 435-445).
Adding selective data is indeed helpful to improve our work. However, as limited by our homemade measurement system, we cannot introduce other gases during the test. How to improve the measurement system and test more data to prove the applicability of our sensor is the focus of our next research.
(2) What is the concentration of hydrogen gas introduced into the gas mix chamber? If the gas is pure hydrogen, please provide its purity.
Response: Thank you very much for the thoughtful suggestion. The hydrogen gas introduced into the gas mixing chamber is a nitrogen-hydrogen mixture with a concentration of 50 ppm. In addition, the pure nitrogen gas introduced into the gas mixing chamber is used to lower the hydrogen concentration, and the purity of the nitrogen gas is 99.999 %. We have added the related information in the revised manuscript.
(3) Figure. 1 in Response to Reviewer is still unclear so that nanoparticles can be found. It just shows yellow spots (maybe corresponding to the intensity from an element). Another clear image directly showing the nanoparticles is required. Otherwise, please explain more details about Figure. 1 and add it into the manuscript.
Response: Thank you for the thoughtful suggestion.
Figure. 1
Figure. 1, shows the EDX mapping result of the synthesized nanostructure. We scanned the area enclosed by the white rectangle and the result is shown in the lower right corner, where the yellow dots represent the palladium element. As can be seen from the picture, a large number of Pd nanoparticles are concentrated in the middle of the scan image, corresponding to the position of the nano fiber in the rectangular region. And only a small fraction of Pd nanoparticles appear in the grayscale region where no nano fibers are present. As a result, the distribution of Pd nanoparticles on the surface of organic nano fibers can be demonstrated. And the related explanation has been added in our revised manuscript.
(4) Regarding Fig.8(b), the authors claim that the relationship between response and hydrogen concentration is exponential. However, it is not compatible with the linear correlation between the response and the square root of hydrogen concentration. In addition, the curve (response vs hydrogen concentration) in Fig.8(b) should be concave downward.
Response: Thank you for the insightful suggestion. It is true that the two relationships are not compatible with each other. For the relationship between response and hydrogen concentration, we are very sorry to have fitted the wrong curve. As explained in previous reply and revised manuscript, the linear correlation between the response and the square root of the hydrogen concentration can be proved and other published articles have reported the similar conclusions. As for the raw data, it may be because the tested hydrogen concentration is concentrated in a lower concentration range, the interval span is not large enough, so there is no well-fitted curve. In view of this, we have modified the corresponding image to visually represented the original data in our revised manuscript.
Figure. 8 (b)

Round 3
Reviewer 3 Report
The revised manuscript is quite improved compared to the previous version. Regarding the selectivity experiment, I can’t agree the reason for not introducing other gases. It may need a small modification with another MFC. However, I accept the authors’ future plan to prove the applicability of the sensor, including selectivity. The authors properly addressed other concerns.